# Soft-Tissue, Rare Earth Element, and Molecular Analyses of *Dreadnoughtus schrani*, an Exceptionally Complete Titanosaur from Argentina

**DOI:** 10.3390/biology11081158

**Published:** 2022-08-02

**Authors:** Elena R. Schroeter, Paul V. Ullmann, Kyle Macauley, Richard D. Ash, Wenxia Zheng, Mary H. Schweitzer, Kenneth J. Lacovara

**Affiliations:** 1Department of Biological Sciences, North Carolina State University, Raleigh, NC 27695, USA; wzheng2@ncsu.edu (W.Z.); mhschwei@ncsu.edu (M.H.S.); 2Department of Geology, Rowan University, Glassboro, NJ 08028, USA; ullmann@rowan.edu (P.V.U.); macauleyk4@students.rowan.edu (K.M.); lacovara@rowan.edu (K.J.L.); 3Department of Geology, University of Maryland, College Park, MD 20742, USA; rdash@umd.edu

**Keywords:** fossil proteins, molecular paleontology, diagenesis, taphonomy, rare earth elements, soft-tissue preservation, geochemistry

## Abstract

**Simple Summary:**

Although many analytical techniques have shown that organic material can be preserved in fossils for millions of years, the geochemical factors that allow this preservation are not well understood. This is partly because paleomolecular studies often do not include geochemical analyses of the fossil or burial environment from which it came. We conducted in-depth geological, geochemical, and molecular analyses of a specimen of *Dreadnoughtus schrani*, an immense dinosaur from Argentina. We reviewed physical aspects of the sediments in which *Dreadnoughtus* was deposited, then characterize the following features: the structural integrity of the bone microstructure; the amount and type of external mineral that infiltrated the bone; the concentration of elements that are rare in the Earth’s crust (REEs) throughout the bone; the preservation of soft-tissue structures (e.g., bone cells and blood vessels); the preservation of bone protein using antibodies that specifically recognize collagen I. Our data show that original bone microstructures and protein are preserved in *Dreadnoughtus*, and that after burial, the specimen was exposed to weakly-oxidizing conditions and groundwaters rich in “light” REEs but experienced little further chemical alteration after this early stage of fossilization. Our findings support the idea that fossils showing lower concentrations of REEs are well suited for molecular analyses.

**Abstract:**

Evidence that organic material preserves in deep time (>1 Ma) has been reported using a wide variety of analytical techniques. However, the comprehensive geochemical data that could aid in building robust hypotheses for how soft-tissues persist over millions of years are lacking from most paleomolecular reports. Here, we analyze the molecular preservation and taphonomic history of the *Dreadnougtus schrani* holotype (MPM-PV 1156) at both macroscopic and microscopic levels. We review the stratigraphy, depositional setting, and physical taphonomy of the *D. schrani* skeletal assemblage, and extensively characterize the preservation and taphonomic history of the humerus at a micro-scale via: (1) histological analysis (structural integrity) and X-ray diffraction (exogenous mineral content); (2) laser ablation-inductively coupled plasma mass spectrometry (analyses of rare earth element content throughout cortex); (3) demineralization and optical microscopy (soft-tissue microstructures); (4) in situ and in-solution immunological assays (presence of endogenous protein). Our data show the *D. schrani* holotype preserves soft-tissue microstructures and remnants of endogenous bone protein. Further, it was exposed to LREE-enriched groundwaters and weakly-oxidizing conditions after burial, but experienced negligible further chemical alteration after early-diagenetic fossilization. These findings support previous hypotheses that fossils that display low trace element uptake are favorable targets for paleomolecular analyses.

## 1. Introduction

In the last few decades, the concept of “exceptional preservation” has expanded beyond specimens that retain exquisite morphological details through processes such as phosphatization (e.g., [1]) to include fossils that preserve some of their original organic content. Evidence that soft-tissue structures, original proteins, and other organic material have been preserved in deep time (>1 Ma) has been reported using a wide variety of analytical techniques, including (but not limited to) amino acid analyses (e.g., [2], Raman spectroscopy (e.g., [3,4]), Fourier transform infrared spectroscopy (FT-IR) (e.g., [5,6,7]), immunology (e.g., [8,9]), time-of-flight secondary ion mass spectrometry (TOF-SIMS) (e.g., [10,11,12]), and tandem mass spectrometry (LC-MS/MS) (e.g., [13,14]). In spite of the diverse, growing literature that original, endogenous organic material can preserve for millions of years, these reports are often regarded with skepticism (e.g., [15]), in no small part, because the geochemical mechanisms that allow for such preservation are not completely understood. Although hypotheses exist as to geochemical factors that may positively influence preservation (e.g., involvement of iron [16,17], microbial activity ([18]), and/or condensation reactions [19,20,21,22]), such studies often examine specific cases in isolation, making it difficult to infer larger scale relationships between the geochemical environment and preservation. As a result, the comprehensive depositional and geochemical data that could aid in building robust, multi-faceted hypotheses about molecular preservation in deep time are lacking from most paleomolecular reports [23,24].

The holotype of *Dreadnoughtus schrani* (MPM-PV 1156) is a specimen that provides the opportunity for in-depth analyses of its preservation and taphonomic history at both the macroscopic and microscopic levels. MPM-PV 1156 represents a massive (59.3 metric tons; 65.4 short tons) titanosaur recovered from the Santa Cruz Province of Argentina [25]. This *D. schrani* holotype has retained 45.3% of its skeleton, including many associated and articulated elements [25]—an extraordinary portion for a sauropod of its size [26]. It has exceptional completeness, coupled with evidence of syndepositional deformation of sedimentary beds, led Lacovara et al. [25] to hypothesize that the individual was entombed during a rapid burial event, such as a crevasse splay. Rapid burial has also been implicated in other types of “exceptional preservation”, including soft-tissues, original organics, and other labile features commonly destroyed by diagenesis (e.g., skin, feathers, cells, proteins) [19,27,28,29,30,31,32,33,34,35]. Given the remarkable preservation of MPM-PV 1156 at the macroscopic level, we hypothesized that the burial event that protected the massive *D. schrani* holotype from extensive biostratinomic processes may have been sufficient to delay early diagenesis at other levels as well, preserving both microstructure and original molecular components, such as proteins, in the fossil tissue.

Here, we review the gross geological context of MPM-PV 1156, including the stratigraphy and depositional sediments of its locality and the physical taphonomy of the *D. schrani* skeletal assemblage as a whole. Then, we perform extensive analyses of the left humerus of the *D. schrani* holotype (MPM-PV 1156-46) to examine its preservation and taphonomic history at a micro-scale. These include: (1) histological analysis and X-ray diffraction (XRD) to assess the structural integrity of the bone microstructure and the extent of its exogenous mineral content; (2) laser ablation inductively coupled plasma mass spectrometry (LA-ICPMS) to assess its geochemical history based on the spatial heterogeneity of rare earth elements (REEs) and other pertinent trace elements throughout the bone cortex; (3) chemical demineralization and optical microscopy of bone tissue to assess the morphological preservation of soft-tissue microstructures; (4) in situ and in-solution immunological assays to assess the presence of endogenous protein.

## 2. Geologic and Taphonomic Context

Specimen MPM-PV 1156 is the holotype of *Dreadnoughtus schrani* [25] (Figure 1A). It was collected between 2005 and 2008 from outcrops of the Late Cretaceous Cerro Fortaleza Formation, along the east bank of the Río La Leona in Santa Cruz Province, Argentina (Figure 1B). Lacovara et al. [25] originally reported its age to be Campanian–Maastrichtian, based on the prior recovery of Campanian ammonites from the underlying La Anita Formation and Maastrichtian palynomorphs from the overlying La Irene Formation (see [36] and references therein). Recent radiometric dating of detrital zircons from the Cerro Fortaleza Formation in the region of the *Dreadnoughtus* quarry constrain the age of these deposits specifically to the Campanian [37]. The 116 skeletal elements of MPM-PV 1156 were found as partially-articulated and closely associated remains alongside those of a second, smaller individual, *D. schrani* paratype MPM-PV 3546 (Figure 2). 

Strata of the Cerro Fortaleza Formation comprise fluvial channel and overbank facies deposited in the Austral Basin, now east of the Southern Patagonian Andes [39,40]. Bones of both *Dreadnoughtus* individuals were recovered from a mixed lithosome comprised of tan, finely trough cross-bedded, fine to medium-grained sandstone and gray, homogenous mudstone containing abundant plant remains. Lacovara et al. [25] interpreted these deposits to represent a crevasse splay horizon deposited onto a fluvial floodplain, based on an abundance of large-scale (primary), convoluted bedforms interpreted to have formed either (1) via liquefaction during rapid deposition, or (2) by sediment redistribution around the large, heavy skeletal remains of the *Dreadnoughtus* holotype at the time of burial, as it subsided into a thick, “soft” substrate. Abundant silicified wood [41] and palynomorphs [42] in strata of the Cerro Fortaleza Formation demonstrate that fluvial floodplains across the region hosted diverse forests. 

Evidence for the authocthonous nature of MPM-PV 1156 includes: (1) most of the tail and the left femur, tibia, and fibula remained in articulation within the south-central portion of the quarry; (2) signs of abrasion to the bones are absent [43,44], and; (3) the enormous size of the individual likely hindered any long-distance transport (Figure 2). Although the bones exhibit negligible weathering [43,44], from the disarticulated nature of large portions of the body (e.g., the ribcage, left forelimb, and dorsal series (Figure 2), we infer the carcass underwent a protracted phase of decay prior to burial. Recovery of several shed cf. *Orkoraptor burkei* teeth within the quarry, along with bite marks on a caudal vertebra of the paratype (MPM-PV 3546) and a dorsal vertebra (belonging to either the holotype or the paratype), further support this conclusion. Preferential preservation of appendicular elements from the left side indicates the animal was likely lying on its left side at the time of burial [25]. 

Based on the quarry map (Figure 2), skeletal element abundances range from 0 to 18 specimens per 4 m^2^ area, with an average of 6 specimens per 4 m^2^. During excavation, multiple bones of both *Dreadnoughtus* individuals were found at strongly plunging angles within the quarry, including the left femur MPM-PV 3546-21 of the paratype, which was found standing vertically (i.e., perpendicular to bedding) [25]. Such plunge angles, in combination with numerous occurrences of bones stacked three deep (e.g., left scapula MPM-PV 1156-48, left radius MPM-PV 1156-51, and dorsal rib MPM-PV 1156-77 near the eastern edge of the quarry; Figure 2), demonstrate that burial occurred in sediment with little bearing strength, in agreement with our prior interpretation of the host sediments deriving from a large-scale crevasse splay event [25]. The general orientations of elongate skeletal elements (e.g., limb bones, ribs, the articulated caudal series) within the quarry form a distinct bimodal pattern, composed of roughly north–south and east–west trends (Figure 2). This might result from the hydraulic alignment of numerous skeletal elements parallel and perpendicular to a paleocurrent flowing along one of these bearings. Individual elements of MPM-PV 1156 range in size from an isolated tooth and distal caudal vertebrae up to girdle elements and stylopodial limb bones (i.e., bones pertaining to both Voorhies Groups I and II were recovered; cf. [45]). All major portions of the body are represented, further supports interpretation of the remains as being autochthonous. Though most of the bones are complete, many exhibit transverse and longitudinal fractures and/or plastic deformation arising from post-fossilization compaction, especially ribs and the pelvic elements; on the left humerus (MPM-PV 1156-49), a distinct fracture through the proximal end is observed (Figure 1C).

## 3. Methods

Brief descriptions of the experimental methods are described below. Comprehensive details of these analyses are provided in the Appendix A, as specified in each section.

### 3.1. Sample Collection

Fossil and sediment samples for molecular assays were collected using aseptic techniques as follows: using nitrile gloves, diaphyseal cortical bone fragments were removed from the humerus of MPM-PV 1156 (Figure 2) immediately upon the opening of its protective field jacket. A sediment sample was taken from within the plaster jacket containing the humerus, from an area between the bone surface and the inner wall of the jacket. Both samples were wrapped in autoclaved foil and stored in an autoclaved glass container under desiccation at room temperature until analyses. During processing, specimens were handled within a fume hood. Bench surfaces were thoroughly cleaned using 95% ethanol, followed by 10% bleach, before sterile foil was laid down over the work area. Nitrile gloves, shoe covers, a hair net, a facemask, and a lab coat (all permanently kept in the ancient-isolated environment) were worn throughout all procedures. For REE analyses, nitrile gloves were used to extract a second fragment of the midshaft of MPM-PV 1156-49 for embedding and sectioning. 

Extant controls for molecular assays, which included extant alligator (*Alligator mississippiensis*) and chicken (*Gallus gallus*) limb bones, were processed separately, and in isolation. All limb elements were separately defleshed, then degreased in 10% Zout (Dial Co., Scottsdale, AZ, USA) or 10% Shout (Johnson Co., New York, NY, USA). Following degreasing, diaphyseal portions of all limb elements were sectioned into pieces, rinsed in Epure water, then wrapped in foil and stored at −80 °C until analyses. The alligator humerus was used for immunofluorescence. All other techniques used the alligator femur. 

### 3.2. Assessments of Preservational Quality and Geochemical History

#### 3.2.1. Histological Analysis

Embedding of *D. schrani* bone tissue samples followed methods described in Green et al. [46], Boyd et al. [47], and Cleland et al. [48]. A fragment of the humerus from MPM-PV 1156 was embedded in Silmar 41^TM^ resin (US Composites, West Palm Beach, FL, USA) and a 2 mm thick, transverse section was cut with a Isomet 1000 precision wafer saw (Buehler, Lake Bluff, IL, USA). The section was then mounted on a glass slide using clear Loctite epoxy (Henkel Adhesives, Rocky Hill, CT, USA) and ground/polished to transparency using a Ecomet 4000 grinder (Buehler, Lake Bluff, IL, USA). Ground sections were imaged with a transmitted light microscope (10x; Axioplan, Zeiss, White Plains, NY, USA) fitted with circularly polarized light filters and a motorized XYZ stage (Marzhauser, Wetzlar, Germany) with output to StereoInvestigator software (MBF Bioscience, Williston, VT, USA) for automated image montaging.

#### 3.2.2. X-ray Diffraction (XRD)

*D. schrani* cortical fragments (~1–2 mg) were powdered in a tungsten carbide Mixer-Mill (SPEX SamplePrep 8000, Cole-Parmer, Vernon Hills, IL, USA) to ~10 µm. Analyses were performed on a X’Pert diffractometer (#DY1738, Philips, Amsterdam, The Netherlands) using Cu Kα radiation (λ = 1.54178 Å), and operating at 45 kV and 40 mA. Diffraction patterns were measured from 5–75° 2θ with a step size of 0.017° 2θ and a time of 1.3 s per step (~0.77 degrees/min). HighScore Plus software version 3.0e (Philips) was used to interpret diffraction traces.

#### 3.2.3. Laser Ablation Inductively Coupled Plasma Mass Spectrometry (LA-ICPMS)

We employed the same mass spectrometry methods as Ullmann et al. [49]. REE concentrations were normalized against the North American Shale Composite (NASC) to enable comparisons to fossil bones from other sites, using values from Gromet et al. [50] and Haskin et al. [51]; a subscript N denotes shale-normalized values or ratios. NASC-normalized REE ratios were used to calculate (Ce/Ce*)_N_, (Ce/Ce**)_N_, (Pr/Pr*)_N_, and (La/La*)_N_ anomalies (sensu [52]) as follows: (Ce/Ce*)_N_ = Ce_N_/(0.5*La_N_ + 0.5*Pr_N_), (Ce/Ce**)_N_ = Ce_N_/(2*Pr_N_ − Nd_N_), (Pr/Pr*)_N_ = Pr_N_/(0.5*Ce_N_ + 0.5*Nd_N_), and (La/La*)_N_ = La_N_/(3*Pr_N_ − 2*Nd_N_). See Appendix A for more details.

### 3.3. Assessments of Soft Tissue Preservation

To prevent exogenous contamination, all sample preparation and analyses performed on fossil materials and all negative controls were conducted using instruments, buffers, and chemicals reserved for and dedicated to fossil analyses, in a laboratory where no extant vertebrate remains were ever housed or analyzed. Preparation and analyses of extant material (i.e., positive controls) occurred in a separate laboratory, and no interchange of solutions, instruments, or other materials occurred. 

#### 3.3.1. Demineralization and Evaluation of Morphological Structures

Fragments of *D. schrani* humerus were incubated in 0.5 M disodium ethylenediaminetetraacetic acid (EDTA) (pH 8.0) for two weeks. Pliable demineralization products were transferred to a glass slide, treated directly with acetone (to rule out all possibility of glue contamination), and imaged in transmitted light and cross-polarized light using a Zeiss Axiocam MRC5 camera mounted to a Zeiss Axioskop 2 Plus biological microscope and a Zeiss Axioskop petrographic polarizing microscope, respectively. Low magnification images were acquired on a Zeiss Stemi 2000-C dissecting microscope. See Appendix A for more details.

#### 3.3.2. Immunofluorescence (IF)

Demineralized tissue from *D. schrani*, chicken, and alligator were embedded in resin, thinly sectioned (220 nm), and subjected to immunofluorescence assays using polyclonal rabbit anti-chicken collagen I antibodies. For detailed descriptions of IF procedures and the numerous antibody specificity tests performed in conjunction with them, see Appendix A.

#### 3.3.3. Enzyme-Linked Immunosorbent Assay (ELISA)

Bone and sediment fragments were demineralized with hydrochloric acid, followed by protein solubilization with guanidine hydrochloride (GuHCl). Buffer samples (empty tubes that received reagent but no tissue; “blanks”) were analyzed simultaneously with fossil and sediment samples to serve as an additional negative control. GuHCl extracts of all tissues and negative controls were subjected to ELISA with polyclonal chicken specific anti-collagen I. For a detailed description of the chemical extraction and ELISA procedures, see Appendix A.

## 4. Results

### 4.1. Preservational Quality

#### 4.1.1. Histological Analysis

The microstructure of the humeral sample from MPM-PV 1156-49 shows primary fibrolamellar bone at the periosteal surface transitioning to densely remodeled secondary bone deeper at the medullary cavity (Figure 3). Structures consistent with fungal tunneling or MDF were not observed in any of the tissue examined, indicating that microbial alteration of this fossil sample was minimal [53]. Under cross-polarized light (Figure 3), deviation from the expected pattern of birefringence in fibrolamellar bone was not apparent, consistent with a lack of alteration of the mineral phase of bone. The lack of indicators of extensive microbial alteration, coupled with the well-preserved microstructure of the observed primary and secondary bone tissue, give this fossil sample an integrity rank of 5 (out of 5) on the Histology Index [54]. 

#### 4.1.2. X-ray Diffraction (XRD)

To determine whether the humeral sample extracted from MPM-PV 1156-49 has been structurally compromised by extensive replacement with exogenous minerals, we performed XRD analysis on a portion of this tissue. Diffraction peaks (Appendix A) recovered in the analysis corresponded with four minerals, including dolomite and three variants of isomorphous [55] apatite (Table 1). Semi-quantitative analysis of the data used gave the approximate percentage of the mineral composition of the fossil tissue as 5% calcite/dolomite and 95% apatite (Table 1), indicating that the replacement and/or inclusion of exogenous minerals was minimal in this sample. 

#### 4.1.3. REE Analyses

LA-ICPMS was used to examine spatial heterogeneity of REE and other pertinent trace elements in the thick bone section. At the whole-bone level, MPM-PV 1156-49 exhibits a ∑REE of 1964 ppm (Table 2). Manganese (Mn; 3233 ppm) and strontium (Sr; 1747 ppm) concentrations are the highest of all recorded elements, with concentrations more than double all the other trace elements examined (Table 2). Yttrium (Y; 505 ppm) is enriched to a similar level as light REE (LREE) elements, whereas the average scandium (Sc) concentration is low (5 ppm). On average, LREE concentrations are an order of magnitude higher than those of the heavy rare earth elements (HREE); the middle rare earths (MREE; Sm–Gd) generally exhibit intermediate concentrations. The average concentrations of uranium (U; 19 ppm) and iron (Fe; 0.19 wt%) are each quite low, owing to low concentrations throughout much of the middle and internal cortices (see below). 

**Intra-Bone REE Depth Profiles**: Concentrations of all REE decline steeply with cortical depth, with profiles for, essentially, every REE forming smoothly decreasing exponential curves (e.g., Figure 4A; see also Appendix A). Among the REE, cerium (Ce) concentrations are the highest at the cortical margin (~4800 ppm) and thulium (Tm) concentrations are the lowest (~10 ppm). LREE exhibit by far the steepest declines in terms of magnitude (on average from ~2300 ppm near the cortical margin to ~100 ppm by 1 cm into the cortex; Figure 4A), generally constituting a decrease of approximately 1.5 orders of magnitude. All MREE concentration profiles exhibit moderate slopes in between those of LREE and HREE. LREE and MREE concentrations are so low throughout much of the middle and internal cortices (<2 ppm) that they frequently encroach on or fall below the lower detection limit (Data S1). In contrast, HREE concentrations generally stay above the detection limit throughout the internal cortex. As exemplified by ytterbium (Yb) in Figure 4A, HREE exhibit the shallowest profiles among the rare earths. The comparatively “shallow” decline in HREE profiles is evident in their magnitude of decrease from the cortical margin: whereas LREE concentrations decrease by two to three orders of magnitude by half way across the width of the cortex, HREE concentrations only decrease by approximately 50% across this same distance (Data S1). 

Even though nearly the entire cortex is composed of densely vascularized Haversian tissue formed by multiple generations of overlapping secondary osteons (Figure 3) [25], there are no obvious spikes in REE concentrations in osteonal tissue surrounding Haversian canals (Figure 4A) nor clear signs of kinks in REE profiles reflective of uptake via double medium diffusion (cf. [56]). Of the elements examined, only iron (Fe) exhibits brief spikes in concentrations in osteonal tissue surrounding a few Haversian canals, primarily within the middle and internal cortices (e.g., at ~15.3 mm; Figure 4C).

Uranium (U) is the only element to exhibit a broad peak in concentrations within the internal cortex (Figure 4B), though concentrations in this region vary substantially. Uranium is also the only element for which concentrations drop to a minimum in the outer portion of the middle cortex. Scandium (Sc) and yttrium (Y) exhibit the same general profile shapes as the MREE in the bone, characterized by a slow, steady decrease in concentrations from the cortical margin (Figure 4C). Strontium (Sr), manganese (Mn), and barium (Ba) each exhibit high and relatively constant concentrations throughout the cortex, with very slightly greater enrichment in the external-most ~3 mm (Figure 4D). 

**NASC-Normalized REE Patterns:** Because concentrations of LREE in the external cortex are considerably higher than through the rest of the cortex, the external 250 μm of the transect exhibits a more LREE-enriched NASC-normalized pattern than the bone as a whole (compare Figure 5B and Figure 6A). Additionally, whereas the external 250 μm plot exhibit a slightly positive Ce anomaly (visually evident as an upward deflection of the pattern at this element), there is none in the spider diagram for the entire bone. However, both plots exhibit a general trend of relative LREE and MREE enrichment relative to HREE. This trend is also evident in how a data point representing the whole-bone composition of MPM-PV 1156-49 plots near the Nd-Gd edge of a Nd_N_-Gd_N_-Yb_N_ ternary plot (Figure 5C). Shale-normalized concentrations range from ~20–100 times NASC values in the external 250 µm of the cortex. 

Plotting each individual laser run compiled into the full transect into a La_N_-Gd_N_-Yb_N_ ternary plot (Figure 5D) shows that there is a tremendous spatial variation in bone composition (i.e., variation greatly exceeds two standard deviations). A spider diagram of individual laser runs (Figure 6B) further confirms this pattern, revealing substantial contrasts in the REE proportions by cortical depth. As these figures show, MPM-PV 1156-49 becomes substantially enriched in HREE relative to LREE and MREE with increasing depth into the cortex. Overall, the bone shifts from being modestly LREE and MREE enriched in the external-most cortex to drastically HREE enriched in the middle and internal cortices (approximately the inner two-thirds of the transect). In the internal-most laser run, HREE are proportionally enriched relative to LREE by over two orders of magnitude (Figure 6B). 

Laser runs across the middle and internal cortices generally exhibit roughly equal depletion in LREE and enrichment in HREE relative to the NASC (Figure 6B). The three most internal laser runs exhibit a slight peak at Nd, and all runs across the middle and internal cortices exhibit a slightly negative Ce anomaly. The internal-most laser run exhibits slightly higher LREE and MREE concentrations than the run immediately external to it. Although there are no clear signs of tetrad effects (i.e., ‘M’- or ‘W’-shaped shale-normalized patterns; [57] and references therein) in individual laser runs (Figure 6B), subtle peaks at Nd, gadolinium (Gd), and holmium (Ho) in the whole-bone and external-most 250 µm spider diagrams (Figure 5B and Figure 6A) may reflect slight tetrad effects (see Appendix A for further discussion of potential tetrad effects in MPM-PV 1156-49). 

**(La/Yb)_N_ vs. (La/Sm)_N_ Ratio Patterns:** At the specimen level, MPM-PV 1156-49 exhibits a high (La/Sm)_N_ value of 3.29 and a (La/Yb)_N_ of 0.88, indicative of substantial LREE enrichment compared to most environmental water samples, dissolved loads, and sedimentary particulates (Figure 7A). This combination of values places the bone just outside the compositional ranges of freshwater in modern rivers and lakes (see Appendix A for the literature sources used for environmental samples). 

When REE ratios are plotted by individual laser runs, the bone exhibits a consistent pattern of decreasing (La/Yb)_N_ and increasing (La/Sm)_N_ as cortical depth progresses (Figure 7B). The laser run including the external margin of the bone exhibits an (La/Yb)_N_ value nearly two orders of magnitude greater than laser runs across the internal cortex. All laser runs across the internal cortex exhibit (La/Yb)_N_ ratios < 0.05, and those across the middle cortex remain <0.4. Laser run (La/Sm)_N_ ratios range roughly from 0.7–10.0. 

**REE Anomalies:** (Ce/Ce*)_N_ and La-corrected (Ce/Ce**)_N_ anomalies are essentially absent in the external-most 5 mm of the cortex (Figure 8A). Values of each of these anomalies, as well as those of (La/La*)_N_, fluctuate both positively and negatively across the transect. Abundant data gaps occur for (Ce/Ce**)_N_ and (La/La*)_N_ anomalies in the middle and internal cortices due to concentrations of praseodymium (Pr) and Nd commonly falling below the lower detection limit and, occasionally, Nd concentrations significantly exceeding those of Pr (Data S1). 

Of all of the anomalies considered, (Ce/Ce*)_N_ exhibits the most variable patterns with cortical depth. Specifically, (Ce/Ce*)_N_ values slowly decrease to a negative peak of ~0.3 in the middle cortex (~16 mm), then steadily rebound to positive values throughout most of the internal cortex (Figure 8A). The highest values recorded along the transect (~9.6) occur near 24 mm, forming the apex of a subtle, positive peak in the outer portion of the internal cortex. However, when averaged across the entire transect (Table 2), these fluctuations cancel each other out; as a whole, MPM-PV 1156-49 exhibits essentially no (Ce/Ce*)_N_ anomaly (value = 0.94). This finding agrees with the lack of any clear inflection of the NASC-normalized pattern at Ce in the whole-bone spider diagram (Figure 5B). 

Following Bau and Dulski [58], (Ce/Ce*)_N_ values were also plotted against (Pr/Pr*)_N_ values in order to differentiate true, redox-related cerium anomalies from apparent anomalies produced by (La/La*)_N_ anomalies. In this plot (Figure 8B), both (Ce/Ce*)_N_ and (Pr/Pr*)_N_ values exhibit significantly greater variation in the middle and internal cortices than is seen in the external-most 1 mm of the bone. This pattern indicates that the internal regions of the bone are relatively more heterogeneous in composition than the external-most portion of the cortex. All but one data point from the external cortex plot near the right margins of fields 3a and 4b, indicative of slightly positive Ce anomalies and variable La anomalies (Figure 8B). The vast majority of data points from the middle and internal cortices plot in fields 2a, 3b, and 4b, generally indicative of positive La anomalies. 

To quantitatively assess these qualitative trends, we also calculated (La/La*)_N_ anomalies and La-corrected (Ce/Ce**)_N_ anomalies (see Methods). (Ce/Ce**)_N_ anomalies are consistently slightly positive throughout most of the external cortex (i.e., the external-most 10 mm exhibits an average anomaly value of 1.40), after which they fluctuate considerably between positive and negative values around an average of zero in the middle cortex (Figure 8A). Unfortunately, minimal data are available from the internal cortex due to concentrations of these LREE often falling below the detection limit. Variations in (Ce/Ce**)_N_ values across the middle cortex encompass a range of more than two orders of magnitude. At the specimen level, the whole-bone exhibits a slightly positive (Ce/Ce**)_N_ anomaly (1.22; Table 2). (Ce/Ce**)_N_ anomaly values are only weakly correlated with U concentrations (r^2^ = 0.54) when they are plotted by the laser run (Figure 8C). (La/La*)_N_ anomalies are commonly positive through the external 17 mm of the bone, after which point data become too sparse to reliably interpret (Figure 8A). This is also evident by the positive (La/La*)_N_ anomaly average for the entire specimen (1.75; Table 2); however, again, this signature is almost entirely derived from the external and middle cortices. 

Yttrium/holmium (Y/Ho) ratios are essentially chondritic (26; [59]) near the cortical margin, but they steadily become increasingly positive with depth (Figure 8A). Y/Ho anomalies form a broad peak in the internal cortex (whose apex is near 23 mm) at values from ~50–600, then gently decline to values from ~60–150 at the internal end of the transect. For the entire specimen (i.e., all transect data averaged together), the Y/Ho anomaly is slightly positive (38; Table 2). 

### 4.2. Soft Tissue Preservation

#### 4.2.1. Demineralization/Morphological Structures

Demineralized cortical bone yielded soft, flexible structures morphologically consistent with vessels, the fibrous collagenous matrix, and osteocytes (Figure 9A–F), and we hereafter refer to them as such for brevity and clarity. During demineralization in ethylenediaminetetraacetic acid (EDTA), vessels were observed emerging directly from the fossil tissue (Figure 9A) as well as in solution. Isolated samples of these hollow, flexible tubes did not dissolve after multiple treatments with acetone, refuting the hypothesis that they are casts formed by the in-filling of acetone-soluble glues and/or field consolidates. The tapering bifurcation pattern observed in modern vessels and in reported soft tissues from other ancient vertebrates (e.g., [60,61]) was also observed among the recovered vessels from *D. schrani* (Figure 9B). These vessels are inconsistent with fungal hyphae, which are cylindrical and grow from an apical tip extension [62,63] and, thus, do not taper when branching. Hyphae are also generally an order of magnitude smaller than the structures we observed (e.g., Figure 2 in [30]). When imaged under cross-polarized light, vessels displayed minimum birefringence, regardless of stage orientation (Figure 9C). Any birefringence that was observed was limited to small, isolated areas along the vessel wall, leaving the majority of the structure dark. Because most minerals are anisotropic and display birefringence under cross-polarized light [64,65], the lack of birefringence in these structures, as well as their pliability, are features more consistent with an amorphous material of organic origin than a mineralized pseudomorph [19,66].

Matrix recovered from *D. schrani* was soft, pliable, and fibrous in appearance, with encased elongated osteocytes oriented with their long axes parallel to one another (Figure 9D). In general, osteocytes displayed shorter, blunted filopodia-like structures in comparison to those observed on putative osteocytes from other extinct taxa [13,60,61,66,67,68]. It is not known if this is a result of tissue degradation during the time between the exhumation of the fossil and analyses, or an artifact of preservation caused by the specific depositional environment in which the specimen was entombed. Regardless, we recovered examples of these elongated structures bearing the hallmark filopodia-like extensions of modern osteocytes (Figure 9E: black arrows). Similar to the vessels, isolated osteocytes did not dissolve in acetone and demonstrated very little birefringence when observed under cross-polarized light (Figure 9F), precluding their origin as either glue or mineral-infill of osteocyte lacunae, respectively [19,66].

#### 4.2.2. Immunofluorescence (IF)

When exposed to commercial polyclonal antibodies raised against chicken collagen I, both chicken (*Gallus gallus*) sections (Figure 10A,D,G,J) and alligator (*Alligator mississippiensis*) sections (Figure 10B,E,H,K) showed a clear signal of immunoreactivity. Binding intensity was greatest in chicken sections (Figure 10D), consistent with our use of antibodies raised against chicken collagen I. Binding was diminished in alligator sections (Figure 10E), which is expected given the phylogenetic distance between alligators and the chicken collagen used as an immunogen [69,70]. Despite the lower signal intensity in alligator, positive binding for this tissue indicates that many of the same collagen epitopes have been retained in both species despite their divergence 237 million years in the past [71]. Thus, we can conclude these antibodies are appropriate for analyses of sauropod bone, as Sauropoda is bracketed phylogenetically by these two extant archosaurs [72] and should, therefore, share the archosaur collagen I epitopes possessed by both *Gallus* and *Alligator*.

Indeed, the collagen I signal was also observed in *D. schrani* tissue sections (Figure 10C,F,I,L), consistent in pattern as with the extant controls (Figure 10F), though the intensity of the signal was greatly reduced relative to both chicken and alligator. Antibody binding was detected within visible sections of tissue, but not in void regions of the sections that contained only resin.

To confirm that the signal observed in *Dreadnoughtus* tissue sections did not result from non-specific binding of the primary antibody, we employed multiple controls. Some sections of each tissue were enzymatically digested with collagenase A (Roche) before incubation with anti-collagen I antibodies [73,74]. This protease specifically cleaves the X-Gly bond in the sequence Pro-X-Gly-Pro, which occurs at a high frequency in collagen but is rare in other proteins [74]. Because collagenase targets a bond abundant in collagen, but rare in non-collagenous proteins to which antibodies may bind non-specifically, collagenase treatment of the tissues in situ provides a specificity control for collagen I antibodies. If the antibodies are binding to collagen, then collagenase will substantially alter the immunoreactive response observed in the tissue; either diminished binding as collagen I is destroyed [75], or increased binding as digestion exposes more epitopes for binding [76,77]. If antibodies are binding non-specifically to proteins other than collagen, then collagenase should have no effect. In both chicken and alligator tissues, the fluorescent signal was greatly reduced after digestion in collagenase for 1 h (Figure 10G,H), indicating that the primary antibody is binding to collagen I specifically, as the targeted destruction of collagen has direct impact on the binding pattern [75]. 

In contrast to the extant controls, digestion for 1 h in collagenase increased the intensity of signal in *D. schrani* tissue (Figure 10I). It has been suggested that condensation reactions, such as Amadori rearrangements and Maillard reactions, may help protect proteins across geologic time through the generation of intermolecular cross-links, forming insoluble aggregates that are resistant to degradation, but which also block reactive epitopes [19,20,21,22]. If such processes have played a role in the preservation of these molecules, brief digestion in collagenase may promote antigen retrieval in fossilized tissues by breaking apart proteinaceous aggregates, exposing more epitopes for binding, and increasing signal intensity [78]. An analogous effect has been observed in extant tissues, as many immunofluorescence protocols optimize the signal by initial incubation of sections in proteinase K, which breaks cross-links generated by formalin during tissue fixation to increase epitope exposure and improve the signal [76]. To test this hypothesis, we extended the duration of collagenase digestion (Figure 11). While sections digested for only 1 h showed a slight increase in antibody binding (Figure 11B), sections incubated in collagenase overnight (Figure 11C) or longer (Figure 11D) demonstrated a near total loss of signal. Additionally, tissue sections themselves became smaller after longer digestions, a further indication they are comprised of material that is directly affected by collagenase. 

In addition to the tissue digestion experiments, we also inhibited the collagen I antibody with an excess of collagen I. In a specific antibody, collagen I blocks all binding sites on the antibody, making them unavailable to bind to epitopes in the tissue [79]. After inhibition, antibody binding to chicken and alligator tissue was completely absent (Figure 10J,K), supporting the specificity of our antibody. *Dreadnoughtus schrani* sections incubated with inhibited antibodies showed a sharp reduction in the fluorescent signal, though inhibition was not as complete as in the extant material (Figure 10L). 

#### 4.2.3. Enzyme-Linked Immunosorbent Assay (ELISA)

ELISA is a method that can identify the presence of protein by quantifying the intensity of color-change generated by antigen–antibody interactions linked to chromogenic substrates [80]. Unlike the in situ immunofluorescence assays, ELISA works by tagging epitopes in chemical extracts of tissue. Because it is performed on chemically extracted antigens that have been solubilized (as opposed to antigen embedded within whole tissue), ELISA is generally an order of magnitude more sensitive than in situ assays [81] and, thus, provides a valuable complement for results obtained through immunofluorescence. Additionally, unlike assays that require extracted proteins to be denatured for analysis (e.g., electrophoresis), ELISA is capable of identifying native epitopes [82]. When incubated with anti-chicken collagen I antibodies, GuHCl extraction products from *D. schrani* tissue showed an absorbance value more than 300% above the value of its secondary-only negative control (Figure 12, columns 1 and 2). This level of absorbance surpasses the standard criteria for a ‘positive’ detection of an antigen, which is twice the background signal (e.g., [83,84,85]). Conversely, extraction products from sediment and buffer controls displayed slightly negative absorbance values, indicating no immunoreactivity was detected in these samples, and that these controls show no evidence of collagen I (Figure 12, columns 3–6). This suggests that the immunoreactivity detected in the *D. schrani* extractions is not the result of contamination derived from the laboratory or burial environments, as these negative controls were subject to the same locations, conditions, and reagents, but contained no reactive material.

The detected signal for *D. schrani* was greatly reduced compared to values obtained for modern controls, which tended to reach saturation (absorbance = 3.0+) within the first 90 min of reading, even with sample concentrations as low as 500 ng/well (Appendix A). After an equivalent amount of time, extraction products from 150 mg of *D. schrani* fossil showed absorbance values approximately 50 times lower than the values obtained for 500 ng of extracted chicken protein. 

## 5. Discussion

The holotype of *Dreadnoughtus schrani* (MPM-PV 1156) represents one of the most morphologically and skeletally complete titanosaur species known [25]. To determine whether the taphonomic conditions that resulted in its extraordinary preservation at the gross morphological level also allowed molecular preservation, we performed extensive analyses of its tissues, including the characterization of its histology, geochemical history, and molecular content. First, the bone microstructure and mineral content was evaluated. Thin sections of the humerus displayed well-preserved primary and secondary bone tissues and lack indicators of microbial attack (fungal tunneling or MDF) or microstructural alteration, giving this specimen a 5/5 on the Histologic Index [54]. XRD analysis showed little exogenous mineralization, with ~95% of the bone mineral comprised of apatite (Appendix A). Such a high level of structural integrity is generally correlated with preserved protein content in archaeological bone [21,86,87,88], indicating that MPM-PV 1156 is an appropriate specimen for more in-depth geochemical and molecular characterization.

### 5.1. Reconstructing the Geochemical History of MPM-PV 1156

Our trace element data elucidate the geochemical history of the *Dreadnoughtus* holotype, as well as clarify the early and late diagenetic conditions to which its left humerus (MPM-PV 1156-49) was exposed, and which allowed it to retain endogenous cells, soft tissues, and collagen I. MPM-PV 1156-49 appears to preserve original, early diagenetic signatures that have not been meaningfully obfuscated by late diagenetic overprinting. This is supported by: (1) a lack of oversteepened elemental profiles that would indicate significant, late uptake (Figure 4); (2) the absence of signs of trace element leaching (i.e., most elemental concentrations increase rather than decrease toward the cortical margin; Figure 4), and; (3) an REE composition most similar in appearance to circum-neutral pH (cf., [89]) rivers and lake freshwaters rather than alkali groundwaters (Figure 7A), which is inconsistent with the incorporation of a major portion of the trace element inventory of the bone from late diagenetic fluids. Overall, the bone exhibits high surface concentrations of LREE (average ~2300 ppm; Data S1), yet its ∑REE value of 1964 ppm is comparable to that of most other Mesozoic bones reported in the prior literature (Table 3), which exhibit ∑REE ranging from ~300 ppm to over 25,000 ppm [90,91,92,93,94,95,96,97,98,99,100], as are its concentrations of Y (505 ppm), Lu (4 ppm), and U (19 ppm). In contrast, Fe (0.19 wt%), Sr (1747 ppm), and Ba (447 ppm) each exhibit low average concentrations compared to other protein-bearing dinosaur bones we have recently analyzed (0.73–1.76 wt%, ~2300–3700 ppm, and ~900–2100 ppm, respectively) [49,100], perhaps reflecting low abundance of these elements within early diagenetic pore fluids at this site. Although these comparisons are rough due to the specimens deriving from diverse taxa and ranging widely in cortical width and depositional circumstances, they demonstrate that the humerus of MPM-PV 1156 generally exhibits average chemical alteration for its age.

Concentrations of REE and Y steeply decline from the cortical margin of the bone (Figure 4A,C), indicating that primary trace element uptake occurred via a single phase of simple diffusion (*sensu* [101]) of one pore fluid. Such concentration depth profiles may arise either via brief uptake from a pore fluid which is not being replenished (e.g., [49]) or as a result of fractionation during protracted uptake from a continually replenished solution (e.g., [102]). We rule out the first of these alternatives due to the significant magnitude of REE enrichment throughout the cortex (e.g., Table 2, Data S1) and heterolithic nature of the entombing sediments, which would have allowed for sustained pore fluid to flow through the remains throughout the early diagenetic, primary phase of trace element uptake. In addition, the cortex of humerus MPM-PV 1156-49 is both extremely thick (~3 cm) and dense—attributes which are known to cause a significant ‘filtering’ effect on trace element diffusion [52,99,100,102,103]. For these reasons, we instead interpret the stark differences in REE enrichment and composition with cortical depth to have arisen via fractionation during protracted, additive uptake (see below). In comparison, the flatter profiles of Fe, Ba, and Mn (Figure 4C,D) suggest that as pore fluids percolated through the bone, these elements were gradually incorporated into homogenously distributed secondary mineral phases, most likely barite, goethite, and Mn oxides [104]. The similarly flat concentration depth profile of Sr is attributed to spatially homogenous substitution for Ca ions in the bone apatite, which commonly occurs during fossilization of bioapatitic tissues [105,106]. 

Elevated concentrations of REE and other trace elements in the external cortex of MPM-PV 1156-49 could reflect either brief uptake from trace element enriched surface and groundwaters (cf., [107]) or protracted uptake from pore fluids possessing low concentrations of these elements (cf., [108]). We find the former of these possibilities unlikely for three reasons. First, most natural waters in fluviodeltaic environments (such as the environment interpreted for MPM-PV 1156 [25]) possess low concentrations of REE, Y, U, and other trace elements [90,109,110,111]. This is because they form a complex with carbonates and humic acids (e.g., [112,113]) and/or may be partially removed in early diagenesis by coprecipitation in secondary phosphates within sediments (e.g., [114]). Second, REE concentration depth profiles within the humerus form typical “simple diffusion” gradients (e.g., La in Figure 4A), which reflect sustained diffusion; they do not show the oversteepened curves that would reflect either preferential uptake from a trace element enriched pore fluid within the external cortex or major late diagenetic uptake (cf., [101]). Finally, many trace element concentrations with relatively slow diffusivities [108], including those of HREE (e.g., Yb), remain >2 ppm throughout most of the internal cortex (see Data S1). This is also consistent with protracted uptake, not brief uptake, because these internal regions are farthest from the external pore fluid source. Thus, we conclude that the bones of MPM-PV 1156 interacted with pore fluids for a longer period of time than other Cretaceous specimens which have been found to retain original protein [49,100]. 

Numerous trace element signatures within the humerus of MPM-PV 1156 demonstrate that the composition of pore fluids percolating through the specimen changed over time through early diagenesis. This is particularly apparent from the spider diagram of REE proportions by transect (Figure 6B), the ternary diagram of REE ratios (Figure 5D), and (La/Yb)_N_ vs. (La/Sm)_N_ plot (Figure 7B), each of which show obvious signs of substantial intra-bone fractionation (*sensu* [52]) occurring during uptake. Specifically, each of these figures show that pore fluids became significantly depleted in LREE by the time they reached the middle and internal cortices. Similarly, despite the similar diffusivities of REE and U in bone [108], they exhibit highly contrasting concentration depth profile shapes (compare Figure 4A,B): REE profiles decline steeply from the cortical margin to low concentrations throughout the interior of the bone, while those of U steadily increase from the internal portion of the middle cortex to a maximum in the internal cortex. Further, the weak correlation between U concentrations and (Ce/Ce*)_N_ values for each laser run (Figure 8C) suggests these were incorporated into the bone over similar timescales [115]. Taken together, it is apparent that the availability of REE diminished as pore fluids percolated deeper into the bone whereas that of U remained high. As discussed by Suarez and Kohn [97] and Kohn and Moses [108], such patterns commonly arise during uptake from oxic fluids, caused by the relatively lower partition coefficients in apatite for U than REE, and greater mobility of U complexes than REE complexes under oxic conditions. Increasingly positive Y/Ho anomalies with cortical depth and positive (La/La*)_N_ anomalies in the external and middle cortices (Figure 8A) are also likely products of fractionation [100]. 

The lack of significant (Ce/Ce*)_N_ or (Ce/Ce**)_N_ anomalies at the whole-bone level (Table 2) and near the cortical margin (Figure 8A) indicates that the early diagenetic environment was neither strongly oxidizing nor reducing. However, slightly negative (Ce/Ce*)_N_ values throughout the middle cortex and the internal portion of the external cortex (Figure 8A) indicate that weak oxidizing conditions prevailed in these regions through the timeframe of uptake. (Ce/Ce**)_N_ anomalies were found to be generally slightly positive in the external cortex, indicative of slightly oxidizing conditions (Figure 8A), which is supported by the presence of high Sc and moderate U concentrations in this region (Figure 4B). Conversely, data points for this region of the bone in the (Ce/Ce*)_N_ vs. (Pr/Pr*)_N_ plot (Figure 8B) fall within fields indicative of slightly reducing conditions. Figure 8A demonstrates that this disagreement arises from the presence of slightly positive (La/La*)_N_ anomalies in the external cortex, which bias calculations of traditional (Ce/Ce*)_N_ anomalies [52]. 

In summary, our cumulative trace element data indicate that the humerus of MPM-PV 1156 experienced protracted trace element uptake from a circum-neutral pH pore fluid during early diagenesis. The similar REE composition of the bone to freshwaters in lakes and rivers indicates that this pore fluid was predominantly fed from surficial sources rather than groundwater sources. By combining these geochemical insights with sedimentologic and taphonomic observations by Lacovara et al. ([25]; i.e., partial articulation, recovery of several cf. *Orkoraptor burkei* teeth within the quarry, burial within a mixed lithosome), we conclude that the carcass of MPM-PV 1156 experienced decay and scavenging for a moderate length of time in close proximity to a fluvial channel on a dry floodplain, after which it became buried by a major crevasse splay event. Its remains were exposed to LREE-enriched groundwaters under weakly-oxidizing conditions for a considerable time following burial, but after early diagenetic fossilization they experienced negligible further chemical alteration.

### 5.2. Evaluating the Preservation of Soft Tissues

The soft, pliable textures of the observed matrix, osteocytes, and vessels are not consistent with mineral in-filling of vessel canals or osteocyte lacunae, and the resistance of these structures to acetone washes precludes glue or consolidate in-filling. All of the osteocytes isolated from the demineralization solution were elongated and flattened in morphology, consistent with osteocytes found in mature bone (i.e., lamellar bone, osteons) [66]. This is consistent with the source fossil, as the humerus of *D. schrani* is composed primarily of remodeled secondary osteons, a trait commonly seen in titanosaurs even before an individual reaches skeletal maturity (e.g., [116,117,118]). Additionally, the vessels recovered are not septate at any point along their length (Figure 9B), distinguishing them from the majority of fungal hyphae [119]. It has been suggested that similar structures previously reported as vessels (e.g., [60]) may represent biofilm endocasts as opposed to original tissues [120]. However, this hypothesis has not been, and is not, supported. Although it has been suggested that biofilms may play a role in endogenous molecular preservation [18], there is no direct evidence that microbes are capable of producing biofilms that can perfectly replicate these structures in fine detail, nor that biofilms are able to retain a three-dimensional shape in solution [67]. Nevertheless, we do not consider morphological similarity, on its own, conclusive evidence that the structures described in this paper are original and endogenous. To that end, we conducted additional experimentation to investigate the endogeneity of the ostensible matrix recovered from *D. schrani*.

Both in situ and in-solution immunological tests supported the preservation of collagen I in *D. schrani* tissues. Immunofluorescent assays displayed in situ antibody binding in fossil sections that was consistent with assays of modern bone (Figure 10). Furthermore, the immunological response of fossil tissues in assays to test for antibody specificity were also consistent with what was observed for modern bone; binding was inhibited when collagen I antibodies were “blocked” by pre-absorption chicken collagen I and was directly affected by the targeted enzymatic digestion of tissue with collagenase. Interestingly, the specific response of the fossil tissue to collagenase is congruous with the presence of diagenetically cross-linked collagen and correlates with an initial increase in signal caused by the exposure of additional epitopes, followed by the subsequent decline in binding as digestion progresses to more extensively degrade the present collagen I. ELISA data showed antibody binding to chemical extraction products that were consistent with a substantial, but not complete, protein loss in fossilized bone compared to recent tissues, and absence of binding in the sediment or laboratory reagents.

Although it has been suggested that immunoreactivity in fossils could be the result of contamination with collagen-like proteins that are produced by bacteria [120] or fungi [121], or non-specific binding to soil microorganisms [122], it has not been explained how these microbial proteins could at once be so ubiquitous as to contaminate specimens from drastically different burial environments and localities, but not ubiquitous enough to be present in the very sediment entombing those specimens, as shown in our ELISA data. Further, the antibody specificity tests conducted on both extant and ancient archosaurian tissue in situ (Figure 9 and Figure 10) support the specificity of the antibody used for ELISA and IF testing to collagen I, indicating that our immunological data are neither the result of cross-reaction with other molecules that may be present in fossilized or non-fossilized bone (digestion control), nor spurious binding with extraneous paratopes in our polyclonal antibody (inhibition control). Thus, the hypothesis that the collagen I signal we have retrieved from *D. schrani* is of exogenous origin is not supported.

Our molecular analyses have demonstrated the presence of soft-tissue and collagen I preservation in the holotype of *Dreadnoughtus schrani* through three independent techniques: (1) morphological identification; (2) in situ localization of antibody–antigen complexes; (3) immunoreactivity to chemical extracts. These assays universally support the identification of collagenous matrix (microscopy), or collagen I specifically (IF, ELISA), and have failed to show evidence of similar molecular content in the entombing sediment or the laboratory environment. Any alternative hypotheses must identify a contaminating agent that contains epitopes recognized by specific antibodies raised against (and inhibited by) archosaur collagen I, which is degraded by collagenase, that is completely absent in the surrounding sediment, and that does not leave histological evidence of microbial/fungal tunneling or destructive foci, and that can contaminate bone tissue both in situ and in-solution without simultaneously contaminating negative control samples that are conducted in tandem. We conclude that the most parsimonious explanation for these results is that original, endogenous collagen I has been preserved in the holotype of *Dreadnoughtus schrani*.

### 5.3. Implications for Future Paleomolecular Studies

That the holotype of *Dreadnoughtus* exhibits average alteration for its age could be viewed as a surprising finding, as most prior reports of endogenous biomolecule recovery from pre-Cenozoic fossil bones derive from specimens exhibiting comparatively less chemical alteration (e.g., *Tyrannosaurus rex* MOR 1125 [100] and *Edmontosaurus* bones from the Standing Rock Hadrosaur Site [49]). However, the cortex sample of MPM-PV 1156-49 found herein to yield original cells, blood vessels, fibrous matrix, and endogenous collagen I consisted primarily of middle and internal cortical tissues, each of which are far less altered than the external cortex of the specimen. Thus, the hypothesis advanced by Trueman et al. [101] and Ullmann et al. [123] that regions of bones exhibiting low uptake of REE are likely the best targets for paleomolecular analyses is supported, even though MPM-PV 1156-49 exhibits ‘average’ alteration at the whole-bone level. The pattern of far greater alteration to the external cortex of MPM-PV 1156-49 than in its more internal regions also conforms to the recommendation by Ullmann et al. [100] for future paleomolecular studies to concentrate sampling efforts on middle and internal cortices rather than the external cortex of fossil specimens. 

Similar to other Cretaceous bones we have analyzed, which are documented to retain endogenous collagen I (see [49,100]), MPM-PV 1156-49 appears to have been preserved under (slightly, in this case) oxidizing conditions (Table 2, Figure 8A). This finding agrees with recent claims by Wiemann et al. [3] and Boatman et al. [7], in that oxidizing depositional environments promote soft tissue and biomolecular preservation (by inducing free radical-mediated molecular condensation reactions). That prolonged early diagenetic exposure to moist conditions did not lead to the complete loss of original organics in MPM-PV 1156-49 is an encouraging finding from the perspective of a molecular paleontologist. In particular, it implies: (1) that other ‘typical’ fossil bones exhibiting ‘average’ levels of alteration might also still retain original biomolecules, and, perhaps more importantly; (2) that ‘normal’ diagenetic pathways to fossilization, such as concurrent recrystallization and permineralization (e.g., [124]), may permit molecular preservation (at least in fluviodeltaic environments). If ‘normal’ bone fossilization processes do not reduce molecular preservation potential to zero, then the pool of fossil specimens that may yield biomolecular material is drastically larger than previously thought (indeed, if this is the case, molecular preservation might not actually be ‘exceptional’). Although recrystallization and permineralization have each been hypothesized to possibly promote molecular preservation in fossil bones (via mineral encapsulation [16,101,125,126,127] and hindrance of microbial infiltration [18,19,128], respectively), it remains premature to claim that ‘average’ fossil bones constitute favorable paleomolecular samples because this outlook remains based on a sample size of one: *Dreadnoughtus* humerus MPM-PV 1156-49. All other protein-bearing, pre-Cenozoic fossil bones whose trace element inventories have been characterized to date exhibit less REE enrichment [49,100], and the REE content of all other specimens documented to yield original molecules (e.g., those analyzed by Tuross [125] and Schweitzer et al. [13]) remain unknown. Therefore, numerous other ‘typically-altered’ fossil bones (e.g., possessing ∑REE > 1500 ppm) must be tested via immunoassays or paleoproteomics to evaluate the true molecular potential of ‘average’ specimens.

## 6. Conclusions

Our assembled molecular and diagenetic data show that, in addition to its exceptional skeletal completeness, the fossil tissue of the *D. schrani* holotype preserved soft-tissue microstructures and remnants of endogenous bone protein. This preservation occurred in a geochemical setting in which its bones were exposed to LREE-enriched groundwaters and weak oxidizing conditions for an extended period after burial. However, following early diagenetic fossilization, the bones experienced negligible further chemical alteration. These findings support the hypotheses advanced by Trueman et al. [101], Ullmann et al. [123], and Gatti et al. [129] that bones exhibiting low trace element uptake are favorable targets for paleomolecular analyses. Moving forward, we encourage the paleomolecular community to include more extensive geochemical analyses as part of their molecular testing routine, as such data will ultimately hold the key to unraveling the complex relationship between diagenesis and ‘exceptional’ preservation.

## Figures and Tables

**Figure 1 biology-11-01158-f001:**
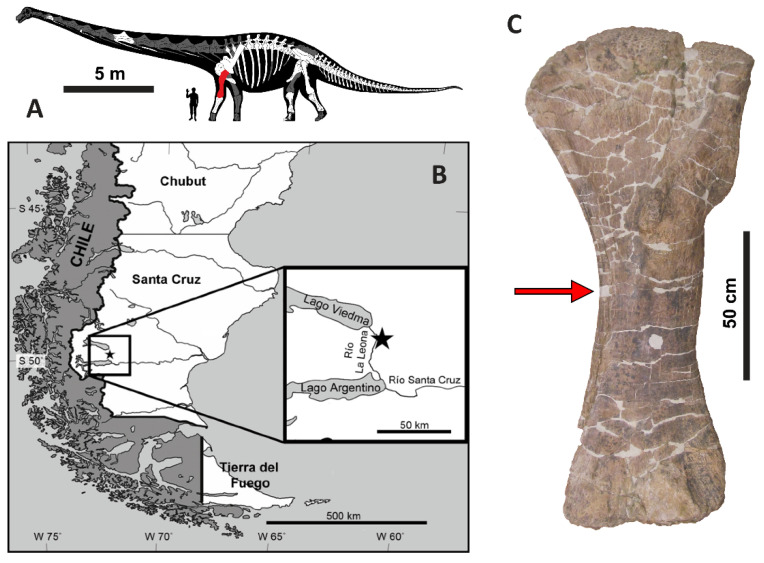
(**A**) Reconstruction of *Dreadnoughtus schrani* holotype (MPM-PV 1156) with preserved elements shown in white. The left humerus (MPM-PV 1156-49; highlighted in red) was examined in this study. (**B**) Map showing the location of the *Dreadnoughtus* quarry in Santa Cruz Province, Patagonia, Argentina. (**C**) Left humerus (MPM-PV 1156-49) of *D. schrani*, shown in anterior view. Red arrow indicates the tissue sampling location for these analyses. Scale bars are as labeled. Image modified from Figures 1–5; of Ullmann and Lacovara [38], with permission.

**Figure 2 biology-11-01158-f002:**
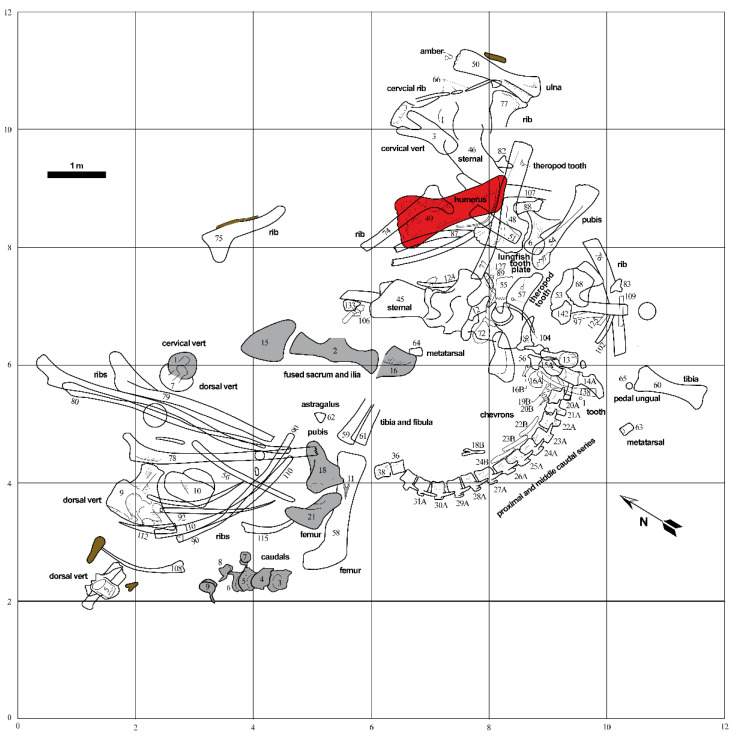
Map of the *Dreadnoughtus schrani* quarry. Numbers denote individual bone specimen numbers. Bones pertaining to paratype individual MPM-PV 3546 are shaded in gray, lignite/wood specimens are shaded in brown, and left humerus (MPM-PV 1156-49) examined in this study is shaded in red. Some overlapping elements and a few specimens found eroding from the outcrop beyond the area depicted in this map are not shown. Select bones are identified as labeled. Grid is shown in 2 m increments; scale bar = 1 m. (Map by J. DiGnazio).

**Figure 3 biology-11-01158-f003:**
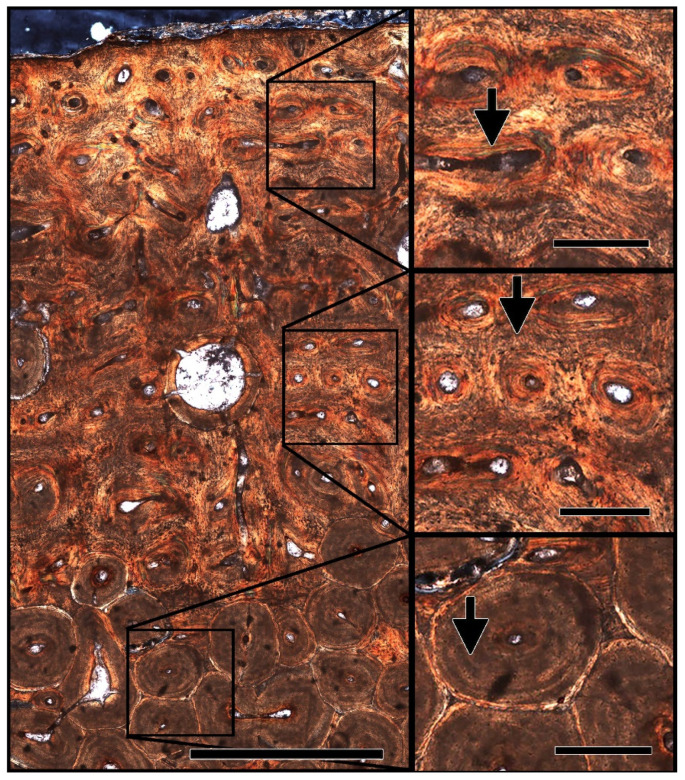
Transverse ground section through a sample of MPM-PV 1156-49, collected from the medial side of the midshaft (complete section is not represented) and imaged under cross-polarized light. The tissue microstructure contains well-preserved primary fibrolamellar bone (FLB) at the periosteal surface, with a birefringence pattern consistent with extant FLB: primary osteons surrounded by concentric layers of lamellar bone (top inset, arrow) and interstitial spaces with unorganized woven bone (middle inset, arrow). Secondary osteons with distinct lamellae (bottom inset, arrow) are also observed. Microscopic destructive foci (MDF) [53] are not apparent, indicating that this sample has not been extensively altered by microbes. Large black spots scattered throughout image represent bubbles in the epoxy adhering the section to the slide. Microstructural preservation of the FLB and Haversian tissues in this fossil sample, and the lack of abundant indicators of microbial attack, suggests that the sample taken from *D. schrani* is an appropriate target for molecular study. Scale bar = 1 mm, inset scale bars = 200 µm.

**Figure 4 biology-11-01158-f004:**
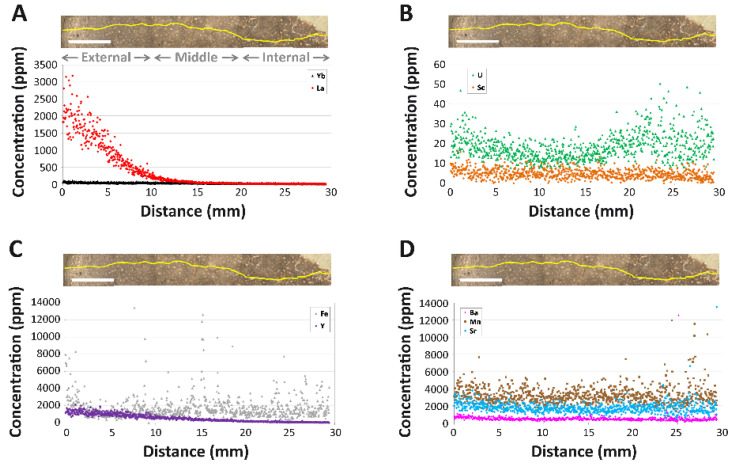
Intra-bone REE concentration gradients of various elements in the left humerus of *Dreadnoughtus* (MPM-PV 1156-49). (**A**) Lanthanum (La) and ytterbium (Yb). (**B**) Scandium (Sc) and uranium (U). (**C**) Iron (Fe) and yttrium (Y). (**D**) Barium (Ba), manganese (Mn), and strontium (Sr). Note that each panel has different concentration scales. Yellow line at the top of each panel depicts the track of the laser across the thick bone section during analyses. Gray text labels in (**A**) indicate the approximate regions of the cortex categorized as ‘external’, ‘middle’, and ‘internal’. Scale bars (in white over bone images) each equal 1 mm.

**Figure 5 biology-11-01158-f005:**
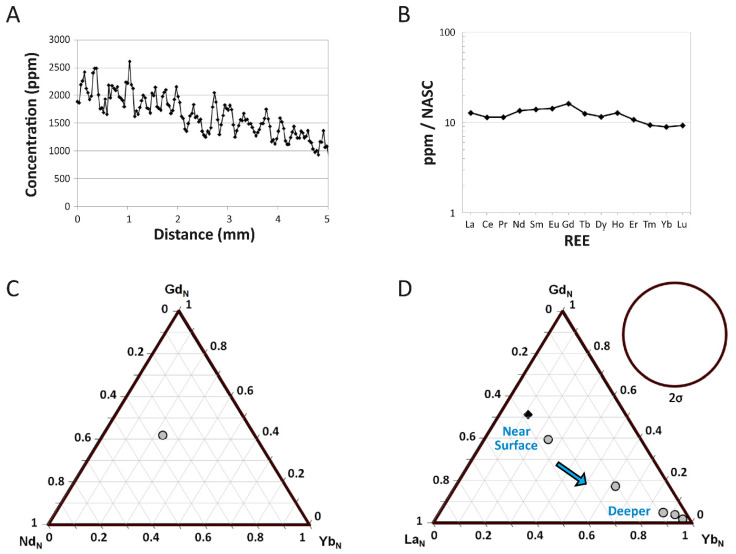
REE composition of the left humerus of *Dreadnoughtus* (MPM-PV 1156-49). (**A**) Three-point moving average profile of La concentrations in the outermost 5 mm of the bone. (**B**) Average NASC-normalized REE composition of the fossil specimen as a whole. (**C**,**D**) Ternary diagrams of NASC-normalized REE. (**C**) Average composition of the bone. (**D**) REE compositions divided into data from each individual laser transect (~5 mm of data each). Compositional data from the transect that included the outer bone edge is denoted by a dark diamond; all other internal transect data are indicated by gray circles. The 2σ circle depicts the area on the plot that represents two standard deviations based on ±5% relative standard deviation.

**Figure 6 biology-11-01158-f006:**
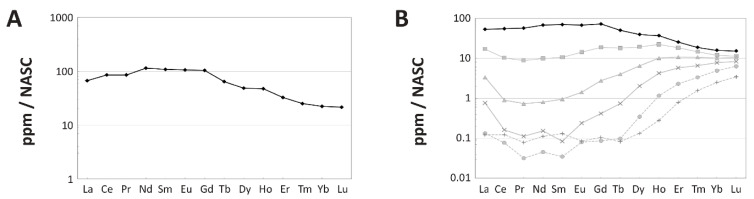
Spider diagrams of intra-bone NASC-normalized REE distribution patterns within the left humerus of *Dreadnoughtus* (MPM-PV 1156-49). (**A**) Average composition of the outermost 250 µm of the cortex, demonstrating a similar degree of relative LREE enrichment in the outermost cortex as seen in the bone as a whole (Figure 5B). (**B**) Variation in compositional patterns by laser transects. The pattern which includes the external margin of the bone is shown in black, those from deepest within the bone by dotted, light-gray lines, and all other analyses in between by solid, dark-gray lines.

**Figure 7 biology-11-01158-f007:**
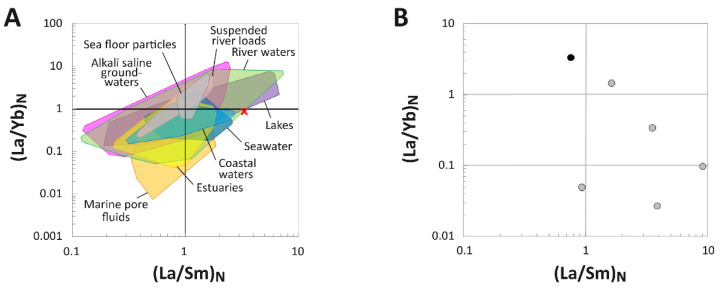
(La/Yb)_N_ and (La/Sm)_N_ ratios of the humerus of *Dreadnoughtus* (MPM-PV 1156-49). (**A**) Comparison of the whole-bone average (La/Yb)_N_ and (La/Sm)_N_ ratios of the fossil to ratios from various environmental waters and sedimentary particulates. Literature sources for environmental samples are provided in the Appendix A. (**B**) REE compositions of individual laser transects expressed as NASC-normalized (La/Yb)_N_ and (La/Sm)_N_ ratios. The transect including the external bone margin is denoted by the black symbol whereas all other (internal) transects are represented by gray symbols.

**Figure 8 biology-11-01158-f008:**
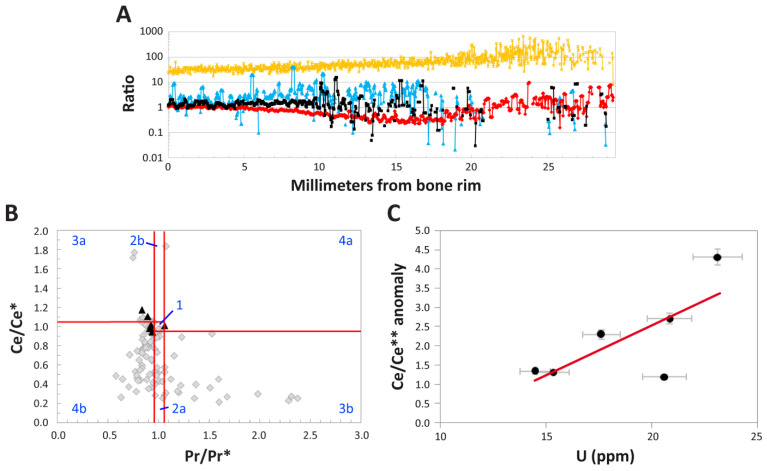
REE anomalies within the left humerus of *Dreadnoughtus* (MPM-PV 1156-49). (**A**) Intra-bone patterns of (Ce/Ce*)_N_ (red curve), (Ce/Ce**)_N_ (black curve), and (La/La*)_N_ (blue curve) anomalies and Y/Ho ratios (orange curve). (**B**) (Ce/Ce*)_N_ vs. (Pr/Pr*)_N_ plot (after Bau and Dulski [58]) of five-point averages along the transect across the cortex of MPM-PV 1156-49 recorded by LA-ICPMS. Separate fields (labeled by blue text) are as follows: 1, neither Ce nor La anomaly; 2a, no Ce and positive La anomaly; 2b, no Ce and negative La anomaly; 3a, positive Ce and negative La anomaly; 3b, negative Ce and positive La anomaly; 4a, negative Ce and negative La anomaly; 4b, positive Ce and positive La anomaly. Measurements from the outer 1 mm of the external cortex are plotted as black triangles, and all measurements from deeper within the bone are plotted as gray diamonds. (**C**) Cerium anomaly (Ce/Ce**)_N_ values plotted against uranium (U) concentrations. Error bars, in gray, are based on analytical reproducibility of ±5%. There is a weak positive correlation (r^2^ = 0.54), shown by the red trendline. Anomaly values were calculated as outlined in the Methods. Absence of (Ce/Ce*)_N_, (Ce/Ce**)_N_, and (La/La*)_N_ anomalies occurs at 1.0.

**Figure 9 biology-11-01158-f009:**
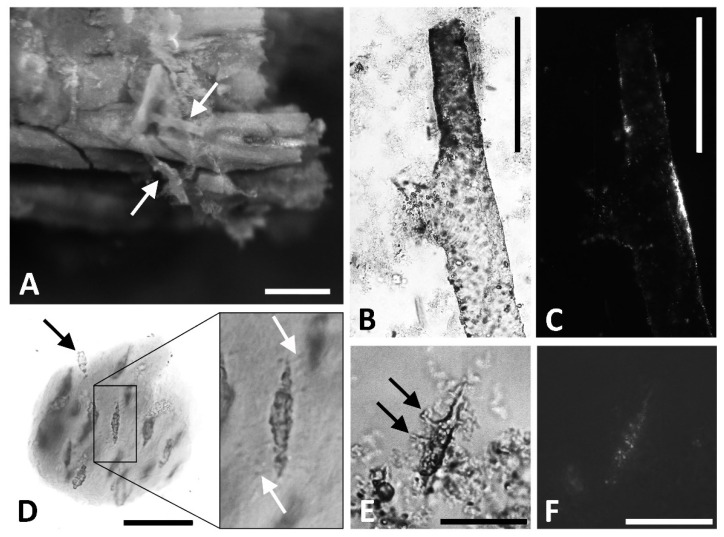
Structures recovered from MPM-PV 1156-49 that are morphologically consistent with soft-tissues. (**A**) White arrows indicate pliable, hollow, transparent tubes emerging from a humeral fragment as the mineral phase is dissolved in EDTA. Scale bar = 500 µm. (**B**) Vessel directly treated three times with acetone, then imaged under transmitted light. Scale bar = 200 µm. (**C**) When imaged under cross-polarized light, the vessel (**B**) showed minimal birefringence, indicating it is not a mineralized structure. Scale bar = 200 µm. (**D**) Fibrous matrix with embedded osteocytes. Osteocytes can be observed emerging from the matrix (black arrow), indicating that these structures are three-dimensional. Inset of the osteocyte shows distinct, lateral projections from the cell-like structure into the surrounding matrix (white arrows), which may represent preserved filopodia or empty canaliculi. Scale bar = 50 µm. (**E**) Isolated osteocyte imaged in transmitted light. Note the narrow, branching structures consistent with filopodia (black arrows). Scale bar = 25 µm. (**F**) Osteocyte imaged under cross-polarized light. Minimal birefringence indicates this structure is not a mineral in-fill of a lacuna. Scale bar = 25 µm.

**Figure 10 biology-11-01158-f010:**
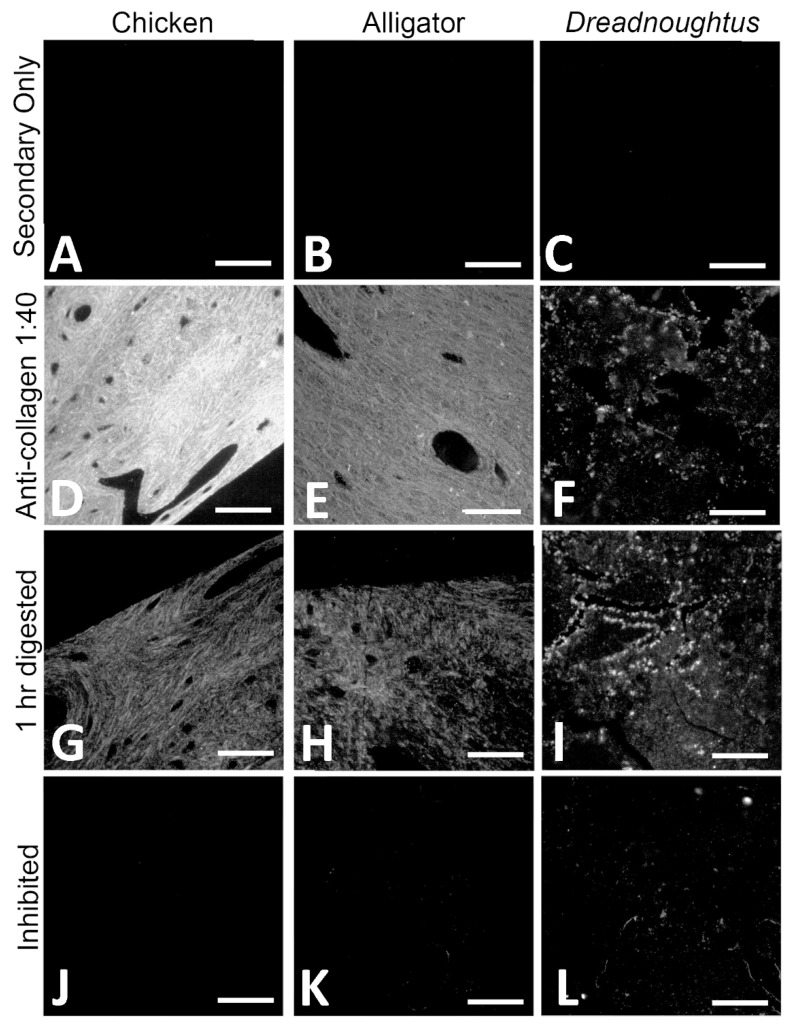
In situ localization of collagen I in demineralized chicken (**A**,**D**,**G**,**J**), alligator (**B**,**E**,**H**,**K**), and *Dreadnoughtus schrani* (**C**,**F**,**I**,**L**) cortical bone fragments using immunofluorescence, imaged at 150 ms. (**A**–**C**). Sections that have been exposed to secondary antibodies without incubation in collagen I antibodies to control for non-specific binding of the secondary antibody and background fluorescence (i.e., “secondary only” control). Lack of signal indicates that the observed fluorescence in other sections cannot be attributed to non-specific binding of the secondary. (**D**–**F**) Sections incubated with anti-collagen I antibodies (diluted 1:40) showed a positive signal for collagen I in all tested specimens. (**G**–**I**) Tissues that have been digested with collagenase prior to exposure to collagen I antibodies show a reduction in signal in the (**G**) chicken and (**H**) alligator sections, and an increase in signal in the (**I**) *D. schrani* section. This indicates that the targeted digestion of collagen I in these tissues directly affect the binding pattern of the antibody, supporting its specificity for collagen I epitopes. (**J**–**L**) Inhibition of the collagen I antibody with chicken collagen shows reduced binding for all taxa, again supporting the specificity of this antibody. Scale bars = 50 µm.

**Figure 11 biology-11-01158-f011:**
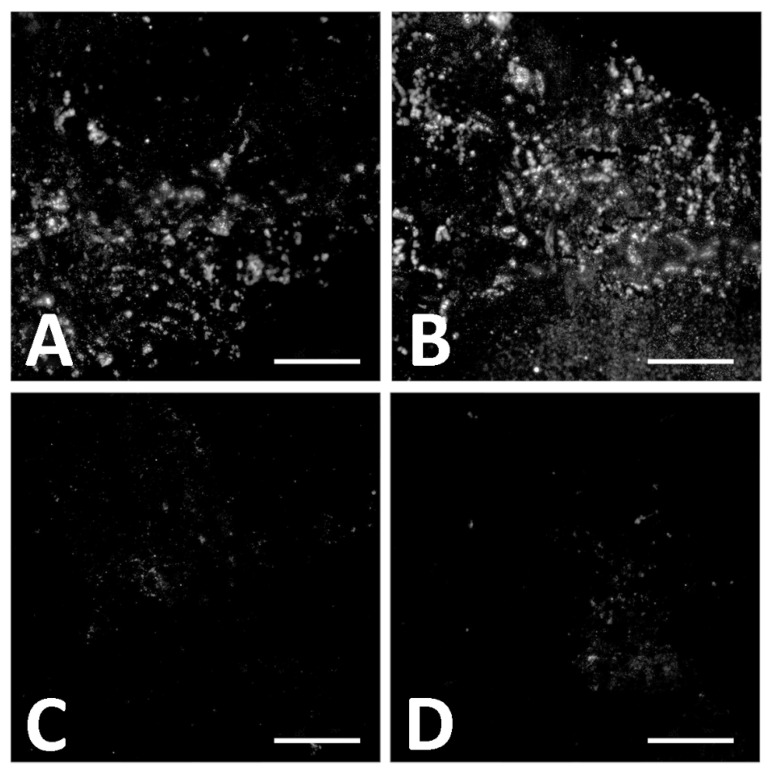
Reduction in immunofluorescent signal in *Dreadnoughtus schrani* tissue with prolonged collagenase digestion prior to incubation with anti-chicken collagen I antibodies. All sections were imaged at an exposure of 150 ms. Sections were digested for (**A**) 0 h (undigested), (**B**) 1 h, (**C**) overnight, and (**D**) 24 h. Compared to the undigested tissue (**A**), immunoreactivity slightly increased with tissue that was digested in collagenase for 1 h prior to incubation with primary antibodies (**B**). This effect is also seen in some extant tissues, as partial digestion in proteinase K increases exposure of epitopes [76]. Extending digestion times to overnight or longer (**C**,**D**) resulted in substantial loss of signal, supporting the specificity of the antibody. Scale bars = 50 µm.

**Figure 12 biology-11-01158-f012:**
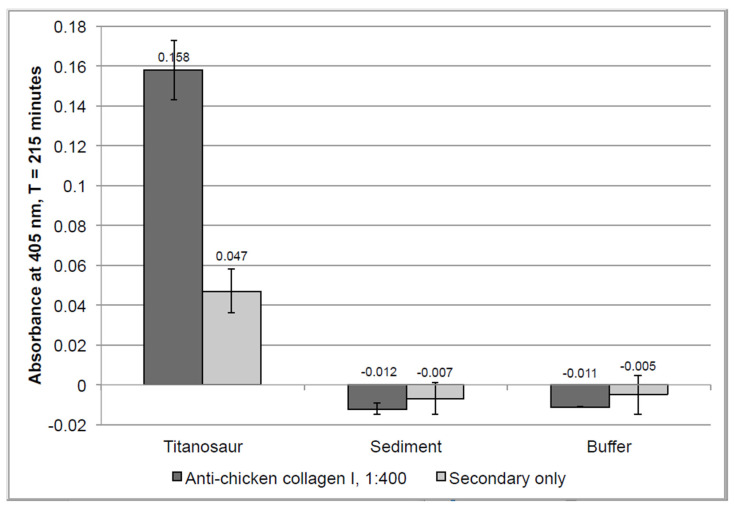
Enzyme−linked immunosorbent assay (ELISA) results of chemical extractions from *Dreadnoughtus schrani* bone, sediment, and laboratory reagents (this trial used the extraction products from 150 mg of bone/sediment per well). Dark gray columns (left) represent absorbance (405 nm, time = 215 min) obtained for extractions incubated with anti-collagen I antibodies (1:400). Light gray columns (right) represent background control (i.e., sample exposed only to secondary antibodies). The absorbance value obtained for *D. schrani* tissue is more than three times its corresponding background control. Negative absorbance values were obtained for extraction products from sediment and buffer samples, corresponding to a lack of immunoreactivity in these samples and indicating that the results obtained for *D. schrani* are not the result of contamination from the entombing environment or lab reagents. Error bars represent one standard deviation above and below mean absorbance values for each sample.

**Table 1 biology-11-01158-t001:** Results of X-ray diffraction (XRD) analysis of the humeral sample from MPM-PV 1156-49. Semi-quantitative analysis of the data (HighScore Plus, Phillips) determined that the mineral composition of the fossil tissue is 95% isomorphous variations of apatite, indicating minimal permineralization with exogenous mineral.

Ref. Code	Score	Mineral	Displacement [°2Th.]	Scale Factor	Chemical Formula	Semi-Quantitative Analysis
01-071-1663	43	Dolomite	0.000	0.334	Mg_0.1_Ca_0.9_CO_3_	5%
01-084-1999	29	Chloroapatite	0.000	0.437	Ca_5_(PO_4_)_3_F_0.09_C_l0.88_	17%
01-071-0880	62	Fluoroapatite	0.000	0.934	Ca_5_(PO_4_)_3_F	41%
01-073-1731	67	Hydroxyapatite	0.000	0.866	Ca_5_(PO_4_)_3_(OH)	36%

**Table 2 biology-11-01158-t002:** Average whole-bone trace element composition of the left humerus of *Dreadnoughtus schrani* (MPM-PV 1156-49). Numbers presented are averages of all transect data acquired across the cortex. Iron (Fe) is presented in weight percent (wt%), all other elements are in parts per million (ppm). Absence of (Ce/Ce*)_N_, (Pr/Pr*)_N_, (Ce/Ce**)_N_, and (La/La*)_N_ anomalies occurs at 1.0. The Y/Ho value reflects this mass ratio.

Element	Concentration
Sc	4.96
Mn	3233
Fe	0.19
Sr	1747
Y	505
Ba	447
Th	0.23
U	18.64
Light Rare Earth Elements (LREEs)	
La	398.95
Ce	762.75
Pr	90.67
Nd	371.52
Middle Rare Earth Elements (MREEs)	
Sm	78.35
Eu	16.91
Gd	84.46
Heavy Rare Earth Elements (HREEs)	
Tb	10.67
Dy	63.76
Ho	13.32
Er	36.67
Tm	4.69
Yb	27.38
Lu	4.23
∑REE	1964
(Ce/Ce*)_N_	0.94
(Pr/Pr*)_N_	0.92
(Ce/Ce**)_N_	1.22
(La/La*)_N_	1.75
Y/Ho	37.90

**Table 3 biology-11-01158-t003:** Summary of the REE composition of the left humerus of *Dreadnoughtus schrani* MPM-PV 1156-49. Qualitative ∑REE content is based on the value shown in Table 2 (1964 ppm) in comparison to values from other Mesozoic bones (as listed in the main text). Abbreviations: DMD, double medium diffusion *sensu* Kohn ([56]); LREE, light rare earth elements.

Clear DMD Kink for LREE?	Relative Noise in Outer Cortex for La	REE Suggest Flow in Marrow Cavity?	Relative ∑REE Content (Whole Bone)	Relative U Content (Whole Bone)	Relative Porosity of the Cortex
No	Low	No	Moderate	Low	Low

## Data Availability

The data presented in this manuscript, as well as extended details on methodology, are available in the Appendix A, which have been made available for download.

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
