# Peer review of "Soft-Tissue, Rare Earth Element, and Molecular Analyses of Dreadnoughtus schrani, an Exceptionally Complete Titanosaur from Argentina"

_biology, 2022, doi:10.3390/biology11081158_

Round 1

Reviewer 1 Report

The manuscript by Schroeter et alii represents an exquisitely crafted example of cutting-edge taphonomic research. Although my experience with paleoproteomics is very limited, and I am not a geochemist, the submitted paper is by all means excellent as far as I can judge.

Some suggestions for minor revisions are as follows:

Abstract: Please change “depositional sediments” to “depositional setting”; “broad taphonomy” to “physical taphonomy”

Keywords: Please consider adding some more keywords to the list.

Intro: For the general readership, you may clarify that the preservation of organic compounds is something different from the preservation of soft proteinaceous structures like horn (e.g, Armitage et al., 2013), baleen (e.g., Gioncada et al., 2016) and hair (e.g., Ji et al., 2002) via e.g. phosphatization or carbonification processes. You may thus consider citing the following works:

• Armitage, M. H., & Anderson, K. L. (2013). Soft sheets of fibrillar bone from a fossil of the supraorbital horn of the dinosaur Triceratops horridus. Acta Histochemica, 115(6), 603-608. https://doi.org/10.1016/j.acthis.2013.01.001

• Gioncada, A., et alii. (2016). Inside baleen: exceptional microstructure preservation in a late Miocene whale skeleton from Peru. Geology, 44(10), 839-842. https://doi.org/10.1130/G38216.1

• Ji, Q., et alii. (2002). The earliest known eutherian mammal. Nature, 416(6883), 816-822. https://doi.org/10.1038/416816a

It is my content that such a specification would prove helpful for many colleagues coming from mostly geological backgrounds!

Line 77 and in several more instances throughout the paper (including figure captions): Please, be sure to put generic and specific names in italics!

Line 120: Lacovara et alii (2014) mentioned possible bite marks affecting i) a vertebra from the paratype and ii) another vertebra that may belong to either the holotype or the paratype. You may slightly rewrite this section to make this point clearer (at present, your text seems to indicate that two vertebrae from the holotype.

Line 805: “cf. Orkoraptor” here vs “cf. Orkoraptor burkei” at line 120: you may consider homogenizing the way you use open nomenclature with respect to these teeth.

Lines 892-895: oxidizing conditions here vs dysoxic conditions elsewhere: this may lead to some confusion, and one may even suggest that soft tissue and biomolecular preservation in the Dreadnoughtus schrani holotype reflects oxygen-depleted (=dysoxic) conditions… For strengthening your point, I would suggest you to slightly expand this part of your Discussion.

Reference #1: Please change “Mccoy” to “McCoy”

Reviewer 2 Report

The authors have produced a valuable, well-documented and convincing manuscript about the taphonomic history of the titanosaur Dreadnoughtus schrani holotype and its feasibility for paleomolecular investigations. The paper provides geochemical data from several different techniques (i.e., XRD, LA-ICP-MS, Immunofluorescence, ELISA, ...) from the left humerus of the specimen. In particular, through the application of these methods and the collection of numerous data, the authors found that D. schrani holotype preserves soft-tissue microstructures and endogenous bone protein.

The manuscript is well organized and well written. Methods are explained in depth and in a quite too long section. The strong point of this paper is that the taphonomic history is well understood. The results shown in this paper will be of interests for taphomomists, biologists and geochemists who would carry out paleomolecular analyses to understand the preservation of soft-tissue structures and organic material such as collagene.

I suggest a minor revision. I have three general comments (1,2,3) and some minor suggestions reported below.

(1) If possible, I suggest to shorten the Methods section, which is now very long, thus lightening the first part of the paper. A solution could be separating the first paragraph about the geo setting, i.e. “Geologic and Taphonomic Context” in a different section (2.) before the section “3. Methods”. In addition, some specific details about the techniques for investigating organic materials could be moved to the Supplementary Material with the other ones.

(2) For the Editor, I would suggest to enlarge some of the figures (e.g., 3, 6, 8, 9, 10) for a better overview of the visual data and diagrams. I suggest also to uniform the figure types making uppercase letters (e.g., A,B,C) for the panels of figures 8, 9 and 10 (now they are lowercase).

(3) Supplementary figures S2, S3, S4 should be included in the text in a unique figure with three panels.

Minor suggestions are reported below:

- line 16: “at both microscopic and macroscopic levels”

- line 17: “depositional setting”

- line 19: “histological observations” or “histological analyses” instead of histology

- line 21: which kind of microscopy? Optical, electronic?...

- In the Keywords, I suggest to add the word “taphonomy”

- line 66: “histological observations” or “histological analyses” or similar, instead of histology

- line 75: I suggest to eliminate the paragraph “2.1. Specimen, Burial Environment, & Sample Acquisition” and to create “2. Geological and Taphonomic Context” separating it from “3. Methods”.

- Please, write in Italics the species and genus name at the following lines: 77, 84, 87, 89, 91, 92, 96, 331, 354, 377, 411, 432, 455, 583, 591, 596, 667, 671, 674.

- line 160: please, add “recent” or “extant” before alligator

- line 168: “Histological analysis/observations” instead of “Histology”

- line 252: “ “ secondary only”). “

- line 385: please add “(Figure 3)” before the citation [24].

- line 620: please, “increased” not in Italics

- line 621: I think there is a mistake in “(Figure 3I)”: maybe it is 9I?

- line 689: “bone microstructure”

- Finally, in the abstract and in the Conclusions, you write “dysoxic conditions”: why they are dysoxic and not oxic as you wrote “oxidizing conditions” in lines 891-895?

Reviewer 3 Report

The submitted MS reports the results of soft-tissue, molecular, and rare earth element analyses of the most morphologically and skeletally complete titanosaur species known. This contribution is more than welcome, because this comprehensive geochemical investigation helps to understand how soft-tissues may persist over millions of years. The presented material has great significance, and only some areas of text had very minor problems (see the general comments and the annotated pdf). Save for some figures (noted on the manuscript), I have no suggestions that might significantly improve the quality of the manuscript or of the accompanying illustrations, but I made some notes (see the main file of the manuscript) that can improve the ease of understanding the text.

I think that this paper is a significant contribution for all people working on REE geochemistry, and also for specialists in taphonomy and palaeobiology of mesozoic vertebrates. It is for me a meritorious work, with relevant results and that deserves to be published. The conclusions are well-supported by the data and the arguments brought by the authors are logical and necessary. The reference list includes papers relevant to the various sections of the manuscript. Other than some very minor corrections (all noted on the main text, in the attached file), I see no need for major interventions on the manuscript, I recommend its publication, and I congratulate the authors for their work and results!

General comments

·       I missed more detailed sedimentological data in the manuscript (e.g., sedimentary section, type of embedding sediment). Also, the cited article did not contain the relevant information needed to describe the former accumulation environment. The MS does not contain stratigraphic section of the site and there is only limited information about the depositional environment! As this work places a strong emphasis on drawing taphonomical conclusions, I would consider it appropriate to extend the sedimentology chapter (more data, more pictures, section etc).

·       I strongly believe that it is necessary to better document the samples used in this work. A figure would be needed (in the text or in the Supplementary data) which show from which part of the bone the specimen originates and what it looks like originally and at different stages of the investigation. Where did the REE geochemical measurements on this sample take place? Where was the histological sample taken from etc.? I think these are of particular importance because the documentation helps to check the results later on….

·       Some figures are only included in the Supplementary data file, although they would be important for understanding the text (result of XRD, Figure S2 and S3 etc) . However, there are some figures in the text that should be moved to the supplementary (e.g. . Figure 2).

·       In some places, especially in lines 708-717, the cited data are a little bit arbitrary and do not cover all useful data. it might be worthwhile to supplement these with all the available data, or not to be so definite

·       Finally, a question which is not a reviewing matter, but I am merely interested in: Why is that after the early diagenetic transformation, the investigated bone experienced negligible further chemical alteration? What “blocked” further REE uptake during bone fossilization? What factors explain the preservation status of this bone?
